# Kinetic Study of CO_2_ Hydration by Small-Molecule Catalysts with A Second Coordination Sphere that Mimic the Effect of the Thr-199 Residue of Carbonic Anhydrase

**DOI:** 10.3390/biomimetics4040066

**Published:** 2019-10-01

**Authors:** DongKook Park, Man Sig Lee

**Affiliations:** Green Materials & Processes Group, Korea Institute of Industrial Technology (KITECH), 55, Jongga-ro, Jung-gu, Ulsan 44413, Korea

**Keywords:** carbonic anhydrase, CO_2_ hydration, Thr-199, kinetics, stopped-flow spectrophotometer

## Abstract

Zinc complexes were synthesized as catalysts that mimic the ability of carbonic anhydrase (CA) for the CO_2_ hydration reaction (H_2_O + CO_2_ → H^+^ + HCO_3_^−^). For these complexes, a tris(2-pyridylmethyl)amine (TPA) ligand mimicking only the active site, and a 6-((bis(pyridin-2-ylmethyl)amino)methyl)pyridin-2-ol (TPA-OH) ligand mimicking the hydrogen-bonding network of the secondary coordination sphere of CA were used. Potentiometric pH titration was used to determine the deprotonation ability of the Zn complexes, and their p*K*_a_ values were found to be 8.0 and 6.8, respectively. Stopped-flow spectrophotometry was used to confirm the CO_2_ hydration rate. The rate constants were measured to be 648.4 and 730.6 M^−1^s^−1^, respectively. The low p*K*_a_ value was attributed to the hydrogen-bonding network of the secondary coordination sphere of the catalyst that mimics the behavior of CA, and this was found to increase the CO_2_ hydration rate of the catalyst.

## 1. Introduction

CO_2_, which is the main cause of global warming, is being actively studied so as to find more economical ways to capture and store this gas stably [1,2,3]. For capturing CO_2_, amine-based chemosorbents, which are expected to be the first material to be commercialized, are currently under investigation [3,4,5,6,7]. However, these chemosorbents have certain drawbacks, in that they require a high amount of heat for regeneration, are corrosive, and undergo deterioration [8,9,10]. Therefore, it is necessary to investigate CO_2_ capture and storage technologies in order to explore new concepts capable of overcoming the limitations of the existing absorbent. Among the various alternative technologies, a newly emerging biomimetic technology that uses the enzyme, carbonic anhydrase (CA), is gaining popularity, because of the rapid reactivity of the enzyme and the environmentally-friendly nature of the process. CA is involved in CO_2_ conversion in living organisms [11,12,13]. The biomimetic technology is a breakthrough, and it is a new concept rather than being an extension of the existing technology. In particular, mineral carbonation using CA is being studied at length for its utilization in the biomineralization of CO_2_ in flue gas, and it has already been undertaken to demonstrate the feasibility of the process in some pilot-scale studies [14,15,16].

CA is a bioinorganic enzyme that contains Zn and catalyzes the reversible hydration of CO_2_ and the dehydration of HCO_3_^−^. The observed rate constant of CA is 10^6^ M^−1^s^−1^, because of which, it is the fastest enzyme known to date [11,17,18]. The structure of CA was determined by X-ray diffraction [19,20,21]. CA has a Zn^2+^ ion at the center of its active site, and is coordinated to the imidazole groups of three different histidine residues (His 96, 94, and 119) and a water molecule or hydroxide ion. The Zn-bound water or hydroxide forms a network with Thr-199 via hydrogen bonding, and Thr-199 forms a hydrogen bond with Glu-106 (Figure 1). X-ray crystallography also revealed the effect of site-specific mutations on the local and global structure of CA. The mutant structures have been used to explain the mechanism of binding and catalysis of the wild-type enzyme [22,23]. Mutations of Thr-199 alter the coordination sphere of zinc, thereby allowing the protein to bind more strongly or more weakly with zinc, as compared with the wild-type enzyme. These substitutions nearly obliterate the CO_2_ hydrase activity, which is consistent with the role of Zn-bound hydroxide as a catalytic nucleophile [23].

The use of CA in the process of capturing CO_2_ is costly, and because of the nature of the enzyme, long-term operation is impossible, thereby complicating the realization of the technology. Therefore, a variety of efforts have been devoted to mimic the structure of CA in order to use it as a CO_2_ hydration catalyst [17,24]. To date, the rate at which CA-model catalysts have been able to achieve the hydration of CO_2_ is 3300 M^−1^s^−1^, which is significantly lower than that of CA. This suggests that it is difficult to obtain a catalyst with a CA-like rate by mimicking only the first coordination sphere of CA. The network of hydrogen bonds in the second coordination sphere is well known to have a great influence on the hydration of CO_2_; thus, it is necessary to develop a CA model catalyst that mimics the second coordination sphere of CA.

The most common assay for CA-mimicking complexes is to measure the esterase activity of model compounds by monitoring the catalyzed hydrolysis of *p*-nitrophenyl acetate (*p*-NPA) [25]. However, this method has a fundamental problem, in that it does not directly measure CO_2_ hydration. We must preserve as many features of the protein as possible in order to successfully mimic the active sites of CA, and to try to synthesize water-soluble molecules to directly measure the activity of CO_2_ hydration [26]. A Zn catalyst with a structure mimicking CA was synthesized; Zn catalyst (**1**) had 4-coordinated tris(2-pyridylmethyl)amine (TPA) as the ligand [27,28,29]. In addition, the ligand 6-((bis(pyridin-2-ylmethyl)amino)methyl)pyridin-2-ol (TPA-OH) was used to synthesize Zn catalyst (**2**), considering the second coordination sphere, including Thr-199 in CA (Figure 2). In this study, we compared the rates of CO_2_ hydration obtained using catalysts (**1**) and (**2**). These catalysts mimic the first coordination sphere of CA and the participation of Thr-199 in the second coordination sphere, respectively. In addition, the p*K*_a_, that is, the intrinsic proton donating ability, and the CO_2_ hydration rate were measured by stopped-flow spectrophotometry, and the results are discussed in detail.

## 2. Materials and Methods

### 2.1. General Consideration

All of the reagents and solvents were purchased from Sigma-Aldrich, and were used without further purification. All of the aqueous solutions were prepared using deionized and distilled water. ^1^H NMR (400 MHz) spectra were recorded on a Bruker AVANCE III instrument. The elemental analyses were measured by a Thermo EA flash 1112, and the mass spectra were obtained with an Agilent 6130. The FT-IR spectra were recorded on a Thermo iS50 FT-IR spectrophotometer, and the potentiometric study was carried out in p*K*_a_ mode using Tiamo 2.3 software, with a Metrohm 808 dosimeter and pH electrode PT1000. The kinetic study was carried out in monochromator mode using an Applied Photophysics SX20 stopped-flow spectrometer equipped with a thermoelectric temperature controller (±0.5 °C).

### 2.2. Materials Synthesis

#### [(TPA)Zn(OH_2_)](ClO_4_)_2_ (1)

A solution of Zn(ClO_4_)_2_·6H_2_O (2.0 mmol, 0.74 g) in acetone (20 mL) was added to a solution of TPA (2.0 mmol, 0.58 g) in acetone (10 mL) under nitrogen. A white precipitate was obtained upon the evaporation of the solution, and consequential washing with diethylether (Yield: ca. 60%). ^1^H NMR (CD_3_OD, 400 MHz) δ = 8.68 (3H, d, pyH), 8.05 (3H, t, pyH), 7.63 (3H, d, pyH), 7.60 (3H, t, pyH; Appendix A). MS(ESI): m/z 372 ((TPA)Zn-OH_2_), 354 ((TPA)Zn), 471 ((TPA)Zn-OH_2_ + ClO_4_), Anal. Calcd for C_18_H_20_Cl_2_N_4_O_9_Zn: C, 37.75; H, 3.52; N, 9.78. Found: C, 37.66; H, 3.42; N, 9.79. 

#### [(TPA-OH)Zn(OH_2_)](ClO_4_)_2_ (2)

The synthesis of 6-methylpyridine-2-yl pivalate (**5**) was as follows: Pivaloyl chloride (**3**) (36 mmol, 4.34 g, 4.43 mL) was added to a solution of 2-hydroxy-6-methylpyridine (**4**) (33 mmol, 3.60 g) in acetonitrile (300 mL), and stirred at room temperature for 2 min. Triethylamine (50 mmol, 6.97 mL) was added and the reaction mixture was stirred at room temperature for 24 h. The mixture was evaporated to dryness, and the resulting slurry was extracted with CH_2_Cl_2_. The organic phase was washed with water, dried over magnesium sulfate, and the solvent was removed by rotary evaporation. The mixture was purified by chromatography on silica gel with CH_2_Cl_2_ as the eluent. (Yellowish oil; Rf value 0.3) ^1^H NMR (CDCl_3_, 400 MHz) δ =7.62 (H, t, pyH), 7.02 (H, d, pyH), 6.80 (H, d, pyH), 2.51 (3H, s, CH_3_), 1.38 (9H, s, CH_3_), ^13^C NMR (CDCl_3_, 400 MHz) δ = 176.77, 158.01, 157.70, 139.39, 121.24, 113.01, 39.05, 27.03, and 24.01 (Figure 3, Appendix A).

The synthesis of 6-(Bromomethyl)pyridine-2-yl pivalate (**6**) was as follows: 6-Methylpyridine-2-yl pivalate (**5**) (2.00 g, 10.20 mol) was dissolved in benzene (450 mL), to which azobisisobutyronitrile (AIBN; 90 mg, approximately 7.5%) was added. The reaction mixture was heated to reflux, and N-bromosuccinimide (NBS; 1.85 g, 10.20 mol) was added in ten fractions every 15 min over 150 min. The reaction mixture was heated under reflux for 7 h, after which it was concentrated to one third of its initial volume and filtered. The solution was collected and evaporated in CH_2_Cl_2_. The desired compound was obtained after chromatography on silica gel with CH_2_Cl_2_ as the eluent, two times. (White solid; Rf value 0.4) ^1^H NMR (CDCl_3_, 400 MHz) δ = 7.78 (H, t, pyH), 7.38 (H, d, pyH), 6.96 (H, d, pyH), 4.51 (2H, s, CH_2_), 1.39 (9H, s, CH_3_), ^13^C NMR (CDCl_3_, 400 MHz) δ = 176.75, 157.78, 156.19, 140.41, 121.55, 115.78, 39.20, 32.89, and 27.06 (Figure 3 and Appendix A).

The synthesis of mono(α-pivalesteropyridylmethyl)bis(2-pyridylmethyl)amine (**8**) was as follows: Bis(2-aminomethylpyridine) (**7**) (0.21 g, 2.06 mmol) was dissolved in acetonitrile (50 mL) and added to a solution of 6-(bromomethyl)pyridine-2yl pivalate (0.29 g, 2.06 mmol) in acetonitrile (100 mL). Sodium carbonate (0.25 g) was added to this mixture, which was heated under reflux for 24 h. The solution was evaporated to dryness. The slurry was extracted with CH_2_Cl_2_ and washed with water (oily solid; Figure 3).

The synthesis of 6-(((pyridin-2-ylmethyl)(pyridin-3-ylmethyl)amino)methyl)pyridin-2-ol (**9**) (TPA-OH) was as follows: mono(α-pivalesteropyridylmethyl)bis(2-pyridylmethyl)amine (**8**) (1.95 g, 5.00 mmol) was placed in a 100-mL round-bottomed flask. To this, 30 mL of 1 M KOH solution (water/methanol, 1/1) was added at room temperature and stirred for 10 h. The reaction was quenched by the addition of water, and the mixture was extracted with CH_2_Cl_2_. The combined organic layers were washed with brine, dried over sodium sulfate, and then concentrated. ^1^H NMR (CD_3_OD, 400 MHz) δ = 11.77 (H, s, OH), 8.66 (2H, d, pyH), 7.67 (2H, t, pyH), 7.35 (2H, d, pyH), 7.25 (2H, t, pyH), 7.21 (1H, t, OH-pyH), 6.44 (1H, d, OH-pyH), 5.95 (1H, d, OH-pyH), 3.95 (4H, s, CH_2_), 3.66 (2H, s, CH_2_), ^13^C NMR (CD_3_OD, 400 MHz) δ = 164.75, 158.06, 148.41, 146.58, 141.98, 137.32, 123.66, 122.56, 117.71, 106.47, 59.81, and 55.58 (Figure 3, Appendix A).

The synthesis of [(TPA-OH)Zn(OH_2_)](ClO_4_)_2_ (**2**) was as follows: TPA-OH (1.46 g, 4.77 mmol) and Zn(ClO_4_)_2_·6H_2_O (1.78 g, 4.77 mmol) were dissolved in methanol (100 mL) and stirred overnight. The desired complex was evaporated to dryness (Appendix A). ^1^H NMR (CD_3_OD, 500 MHz) δ =8.87 (2H, d, pyH), 8.08 (2H, t, pyH), 7.83 (1H, t, OH-pyH), 7.64 (2H, t, pyH), 7.60 (2H, d, pyH), 7.00 (1H, d, OH-pyH), 6.81 (1H, t, OH-pyH), 4.35 (4H, s, CH_2_), 4.20 (2H, s, CH_2_), ^13^C NMR (CD_3_OD, 400 MHz) δ = 164.88, 155.10, 151.80, 147.89, 143.48, 141.12, 124.94, 124.59, 115.37, 111.46, 57.95, and 57.72, (Appendix A). MS(ESI): m/z 388 ((TPA-OH)Zn-OH_2_), 370 ((TPA-OH)Zn).

### 2.3. Potentiometric pH Titration

The electrode system was calibrated with a Metrohm standard buffer solution with a pH of 4.00, 7.00, and 9.00, before titration. An aqueous solution of each of the complexes (1.0 mM) was dissolved in 2.0 mM of HNO_3_ (I = 0.1 M NaNO_3_) and stirred at 25 °C. Titrations were performed with standardized 0.1 M NaOH solutions, while the pH was monitored to identify the half-equivalence points using Tiamo 2.3 software.

### 2.4. Kinetic Measurements (Stopped-Flow Spectrophotometer)

The measurements of the CO_2_ hydration rate were carried out by a pH indicator method using a stopped-flow spectrophotometer, similar to those previously described [30,31,32]. The pH time dependence by CO_2_ hydration was observed as the change in the absorbance of the indicator In^−^ to HIn, and the CO_2_ hydration rate constant, *k*_obs_, was determined by Equation (1). The Zn-bound HCO_3_^−^ was rapidly replaced by H_2_O to regenerate L-Zn-OH_2_.
(1)$$[L\text{-}\mathrm{Zn}\text{-}{\mathrm{OH}}_{2}{]}^{2+}\;+\;{\mathrm{CO}}_{2}\;+\;{\mathrm{In}}^{-}\;\overset{k_{obs}}{\to }\;[L\text{-}\mathrm{Zn}\text{-}{\mathrm{OCO}}_{2}H{]}^{+}\;+\;\mathrm{HIn}$$

Prior to the experiment, a solution of CO_2_ saturated water was prepared by purging deionized water with 100% CO_2_ gas at 25 °C for at least 1 h. Using Henry’s constant, this solution was calculated to contain 33.8 mM CO_2_. Additionally, a solution containing 0.2 M NaClO_4_; 0.1 M *N*-(1,1-dimethyl-2-hydroxyethyl)-3-amino-2-hydroxypropanesulfonic acid (AMPSO) buffer; and 5 × 10^−5^ M thymol blue indicator, pH 9.0 (adjusted using NaOH), was evacuated under a vacuum for 1 h, followed by purging with nitrogen for 30 min to remove dissolved CO_2_. The baseline uncatalyzed rate was found by rapidly mixing the dissolved CO_2_ solution and the buffer solution (1:1) in the stopped-flow spectrophotometer, while recording the time-dependent absorbance at λ = 596 nm. Twenty milliliters of 1 mM solutions of (**1**) and (**2**) in the AMPSO buffer solution were prepared by stirring under nitrogen. The catalyst concentration-dependent initial rates were found by diluting the 1.0 mM solutions with the AMPSO buffer solution. The initial rates for each catalyst concentration were calculated by fitting the first 10% of the time-dependent absorbance data with a single exponential decay function. All of the kinetic runs were performed for 100 s in order to ensure that equilibrium was reached and that the final absorbance value was indicative of reaction completion. Therefore, the time period for fitting the initial rates could be accurately defined for each reaction. Each concentration of catalyst was measured four times, and the standard deviation was determined. The initial rate of CO_2_ hydration is defined by Equation (2), where x indicates the concentration of In^−^ in Equation (1). The change in absorbance with time by In^−^ can be represented by Equation (2).
(2)$$V_{\mathrm{init}}\;=\;{\left(\frac{\mathrm{dx}}{\mathrm{dt}}\right)}_{t\to 0}\;=\;Q{\left(\frac{\mathrm{dA}}{\mathrm{dt}}\right)}_{t\to 0}=\;Q\left(A_{0}-A_{e}\right){\left[\frac{d\left(\ln\left(A-A_{e}\right)\right)}{\mathrm{dt}}\right]}_{t\to 0}$$
where Q is the buffer factor, which was determined experimentally by titrating the AMPSO buffer solution with three concentrations of HCl in the stopped-flow spectrophotometer, and A_0_ and A_e_ are the initial and final absorbance values, respectively. The measured HCl concentrations were 7.6, 15.3, and 31.0 mM, and were chosen to represent a range of [H^+^], similar to the [H^+^] generated in the CO_2_ hydration reaction [30]. The second-order rate constant, *k*_obs_, for Equation (1) is defined by Equation (3), and the error is calculated as the deviation of the curve from linearity.

(3)$$V_{init}=\;k_{obs}\left[{\mathrm{Zn}}_{\mathrm{tot}}\right]\left[{\mathrm{CO}}_{2}\right]$$

## 3. Results and Discussion

### 3.1. General Consideration

Designing an effective catalyst structure requires an in-depth understanding of the mechanism and the factors responsible for controlling the target reaction. The mechanism of action of CA involves three stages. The first stage is a CO_2_ addition reaction, wherein the Zn-bound hydroxide reacts with CO_2_ to form a Zn-bound bicarbonate [11,33]. The bicarbonate formed in the second step is released by the substitution reaction of the solvent water and bicarbonate. The final step entails the regeneration of the catalyst in its active form, that is, the deprotonation of the Zn-bound water to form the Zn-bound hydroxide (p*K*_a_ = 7; Figure 4). Among these steps, deprotonation is known to be the rate-determining step (RDS) [30,34,35,36].

The structure of natural CA includes a tridentate N3 ligand coordinated to and surrounding the Zn ion. However, studies of catalysts mimicking CA reported thus far show that the complex with the tetradentate N4 ligand has a higher activity than that with the tridentate N3 ligand [24,30]. According to Koziol et al., tetradentate N4 ligands form weaker bonds with the Zn-bound hydroxide ion than do tridentate N3 ligands. Furthermore, N3 ligands prefer a bidentate mode for the binding of Zn-bound bicarbonate, while N4 ligands prefer a monodentate mode [30]. This is because the Zn-bound bicarbonate of the N4 ligand forms weaker bonds than the N3 ligand, so that the substitution reaction of the water molecules proceeds easily. In this study, the catalyst we designed to mimic CA was synthesized using TPA, which contains pyridine, as the N4 coordinating ligand. In addition, the water molecule or hydroxide ion coordinated to the Zn ion in CA was designed to form part of an intermolecular hydrogen-bonded network, such as that containing Thr-199. The proton at position two of the pyridine unit of TPA was substituted with a hydroxyl group, which formed a hydrogen bond with the Zn-bound water, to ultimately form a network of hydrogen bonds (Figure 2). The formation of hexagonal rings as a result of intramolecular hydrogen bonding would be expected to stabilize the structure. The substitution of the proton at position three of the pyridine unit with a hydroxyl substituent would make it impossible for the hydrogen-bonded network to form, because the distance between this substituent and the hydroxide ion or water molecule coordinated to Zn would be too large. Therefore, a Zn complex was synthesized by substituting the hydroxyl group at position two of the pyridine unit of TPA. The environment surrounding the water molecules bound to (**1**) and (**2**) was investigated by studying the O–H vibration using FTIR spectroscopy. (Appendix A). All of the complexes exhibited a sharp band corresponding to the O–H bending vibration at approximately 1610 cm^−1^ [37]. In particular, (**1**) showed bands at 3254 and 3336 cm^−1^, corresponding to the O–H stretching mode, while (**2**) showed a broad peak around 3488 cm^−1^, attributable to hydrogen bonding [38,39]

### 3.2. pK_a_ Measurement

The deprotonation of the Zn-bound water in the CO_2_ hydration reaction of CA is a crucial step in determining the reaction rate, as it is the rate-determining step [30,34,40]. Therefore, the deprotonation ability (i.e., the acidity of water molecules) in CA-mimicking catalysts is a fundamental parameter for determining the ease with which the nucleophilic hydroxide ions can be generated at a neutral pH. The acidity of compounds (**1**) and (**2**) was measured using potentiometric pH titration, and the results are provided in Table 1.

The p*K*_a_ of (**1**) was found to be 8.0 (8.03), as previously reported, while that of (**2**) was found to be 6.8 (6.80), which is same as the CA value [28,41]. In the case of (**2**), the introduction of the hydroxyl substituent onto the pyridine unit can be expected to have both an electronic and a structural effect. The ^1^H NMR spectra in Appendix A reveal that the peaks due to pyridine with hydroxyl groups shifted upfield, as compared to those of the pyridine. In particular, the peaks attributed to N-CH_2_- showed a chemical shift from 4.35 ppm to 4.20 ppm upon the introduction of a hydroxyl group (Appendix A). These results indicated that the introduction of hydroxyl groups donates a sufficient number of electrons to the Zn ions. This is expected to reduce the acidity of the metal and consequently increase the p*K*_a_ of the coordinated water molecule [40]. However, our experimental results were contradictory to this expectation, and the p*K*_a_ was found to decrease. Therefore, it was concluded that the structural effect had a more significant impact on the p*K*_a_. The deprotonated structure of (**2**) can be deduced from that of CA [42,43]. It can be deduced that the O atom of the hydroxyl substituent at position two of the pyridine unit and the H atom of the Zn-bound hydroxide ion form a hydrogen bond to stabilize the structure. The formation of this hydrogen bond creates a six-membered ring that includes the Zn-bound hydroxide, such that the deprotonated water is stabilized (Figure 2). Juan et al. reported the crystal structure of Zn-bound hydroxide, using a ligand that introduced an amino group at the two-position of TPA [28]. The amino group and Zn-bound hydroxide formed a hydrogen bond network and stabilized through the distorted hexagonal ring structure. They also observed that p*K*_a_ was lowered up to 6.0 upon the introduction of amino groups.

Krebs et al. investigated the importance of a conserved hydrogen bonding network through amino acid substitutions (Ser, Ala, Val, and Pro) at Thr-199 of CA [44]. Mutation of Thr-199 -> Ser, which is capable of forming a hydrogen bond with Zn-bound water, showed little increase in p*K*_a_. However, mutations to Ala, Val, and Pro, which have aliphatic side chains, increased the p*K*_a_ by 1.5 to 2.5 units. Therefore, as in the case of (**2**), the hydrogen bond network of Zn-bound water is a very important factor in controlling the p*K*_a_, and it is necessary to consider the structure of the secondary coordination sphere in the designing of CA-mimicking complexes.

### 3.3. Kinetic Measurements

As the deprotonation step in the CO_2_ hydration reaction of CA is known as the rate-determining step, the catalyst was designed to lower the p*K*_a_ value [30,40]. To summarize the aforementioned results, it was experimentally confirmed that the lower the p*K*_a_, the faster the CO_2_ hydration reaction occurs. Our experimental results show that (**2**) has a p*K*_a_ value similar to that of CA. In addition, it is necessary to further investigate the rate changes caused by the hydrogen-bonding network, including Thr-199.

To understand the activity of the CO_2_ hydration reaction of (**1**) and (**2**), the rate constant (*k*_obs_) of the reaction was measured using a stopped-flow spectrophotometer (Appendix A). The fast-kinetic measurement method was used, because the CO_2_ hydration reaction catalyzed by the CA mimic was expected to be very fast. Thus, *k*_obs_ was determined from the initial value obtained by the single exponential decay fitting of the absorbance change, corresponding to 10% of the total reaction. The results are listed in Table 2. As expected, the catalytic activity of (**2**), with a lower p*K*_a_ value, is 13% faster than that of (**1**).

The species that are directly active in the CO_2_ hydration reaction are deprotonated, [(TPA)Zn(OH^−^)] and [(TPA-OH)Zn(OH^−^)]. The distribution ratio of the deprotonated species in (**1**) and (**2**) is obtained by the acid–base equilibrium determined by the p*K*_a_ value, and *k*_obs_ can be expressed as *k_ind_* (pH-independent form; Equations (4) and (5), Appendix A).

(4)
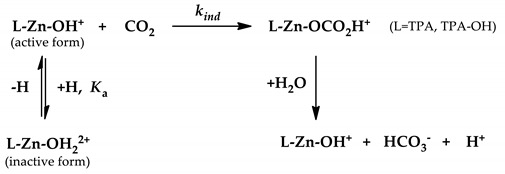


(5)$$V_{ind}=\;k_{ind}\left[{\mathrm{LZnOH}}^{-}\right]\left[{\mathrm{CO}}_{2}\right]$$

Interestingly, *k*_ind_ also showed a rate constant of (**2**) greater than (**1**). This result indicates that (**2**) is more efficient than (**1**) in the hydration reaction through the nucleophilic attack of Zn-bound hydroxide (active form) on CO_2_. Many researchers have attempted to lower the p*K*_a_ of Zn-bound water by increasing the Lewis acidity of Zn^2+^, using ligands that attract more electrons [17,28,30,36]. However, increasing the acidity of Zn^2+^ increases the binding force for the Zn-bound bicarbonate, which is not effective for the release of bicarbonate. According to Koziol et al., ligands that donate more electrons form weak Lewis acids in Zn^2+^, thus producing a greater amount of free hydroxide and bicarbonate ions [30]. The rate of substitution of water molecules is expected to increase, because bicarbonate ions weakly bound to Zn can be replaced by water more easily. Indeed, in many studies that mimic CA, N4 ligands have bene demonstrated to show a higher activity than N3 ligands, because the binding of bicarbonate is weakened by the donation of more electrons. In our recent study, mimic CAs were synthesized using Ni^2+^ and Cu^2+^ instead of Zn^2+^, to increase their acidity, and the p*K*_a_ values were measured to be 6.0 and 7.6, respectively [32]. Despite the low p*K*_a_ values, the *k*_ind_ values of the Ni^2+^ and Cu^2+^ compounds were 542.3 and 526.4 M^−1^s^−1^, respectively, and the rate constant was significantly lower than that of the Zn^2+^ complex (710.3 M^−1^s^−1^). That is, in the Ni and Cu compounds, the central metal has a higher acidity than that of the Zn complex, and thus, they form strong bonds with bicarbonate, making it difficult to substitute water molecules. Looking closely at the formation mechanism of the bicarbonate intermediate of CA, as reported by Lindskog, Zn^2+^ is capable of monodentate as well as bidentate bonding with bicarbonate (Figure 5) [33,45]. Increasing the acidity of Zn increases the binding force between Zn and the bicarbonate, and makes the conversion of bidentate to monodetate bonds difficult. Compound (**2**) will lower the acidity of Zn^2+^ upon the introduction of the hydroxyl group as the electron donating group in pyridine. Thus, Zn^2 +^ of (**2**) rather than (**1**) shows that weak bonds with bicarbonate are formed, increasing the substitution rate of water molecules.

Juan et al. reported that the p*K*_a_ was lowered to 6.0 upon replacing up to three amino groups at the two-position of TPA [28]. Their argument was that the hydrogen bonding network around the Zn-bound water can stabilize the Zn-bound hydroxide. Through crystal structure analysis, they demonstrated that the hydrogen bonding network increases the bond length of the Zn-bound hydroxide, and this also increases the nucleophilicity of hydroxide. Our study also revealed that the p*K*_a_ was lowered by the formation of a hydrogen bonding network through the introduction of hydroxyl groups. The hydroxyl group introduced into TPA can lower the acidity of Zn^2+^; this is an interesting result, in that the bond between Zn^2+^ and the hydroxide is weakened, and the bicarbonate produced through nucleophilic attack at CO_2_ is easily released (Figure 6).

## 4. Conclusions

We synthesized a Zn catalyst that mimics the role of Thr-199 in the second coordination sphere of CA as well as its active site. For the mimicking Thr-199, a hydroxyl group was introduced at the two-position of PTA, which showed that the p*K*_a_ value was lowered by forming a hydrogen bonding network around the Zn-bound water. Unlike the conventional strategy of decreasing the p*K*_a_ value by increasing the acidity of Zn, the acidity of Zn could be decreased through the hydrogen bond network, and the p*K*_a_ value could also be lowered. The decrease in Zn acidity was observed to reduce the affinity between the Zn ions and bicarbonate, resulting in the effective release of bicarbonate, increasing the catalyst cycle rate. We successfully synthesized a mimic catalyst with better conversion efficiency by introducing hydrogen bonds into the second coordination sphere as well as the active site of CA. This result suggests that catalysts with a better conversion efficiency can be synthesized by designing various functions for the second coordination sphere in CA.

## Figures and Tables

**Figure 1 biomimetics-04-00066-f001:**
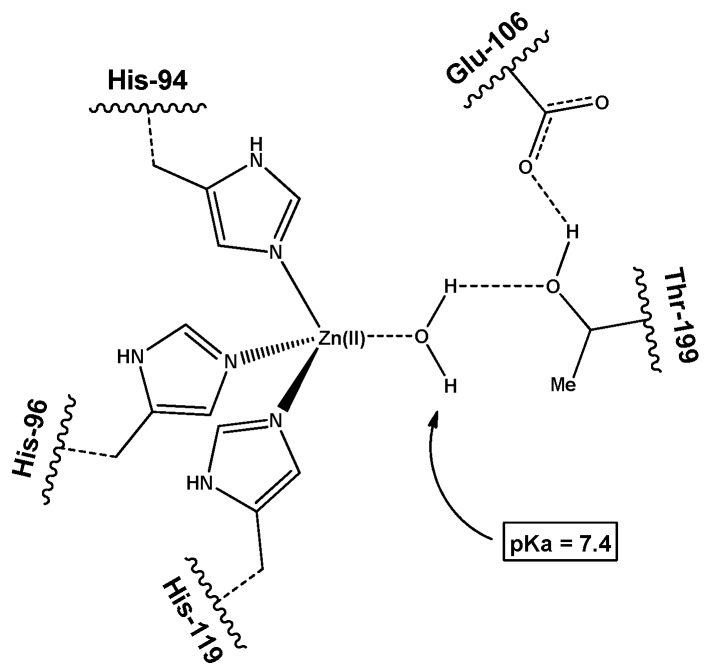
Zn-bound water and nonbonded interactions of Thr-199 and Glu-106 residues.

**Figure 2 biomimetics-04-00066-f002:**
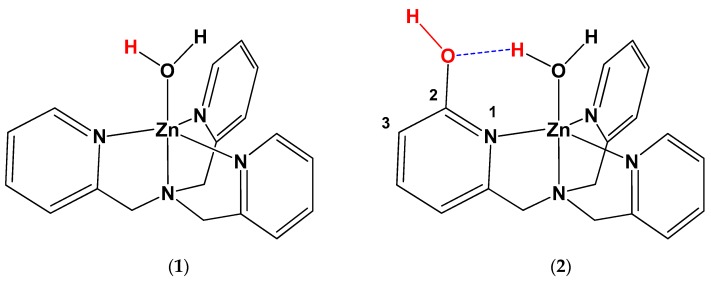
Structures of the two carbonic anhydrase (CA)-mimicking Zn catalysts, namely: (**1**) (tris(2-pyridylmethyl)amine)Zn(OH_2_) complex and (**2**) (6-((bis(pyridin-2-ylmethyl)amino)methyl)pyridin-2-ol)Zn(OH_2_) complex.

**Figure 3 biomimetics-04-00066-f003:**
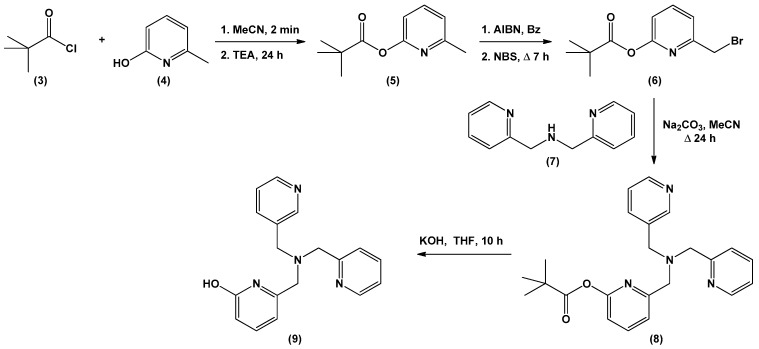
Synthesis of 6-(((pyridin-2-ylmethyl)(pyridin-3-ylmethyl)amino)methyl)pyridin-2-ol (**9**).

**Figure 4 biomimetics-04-00066-f004:**
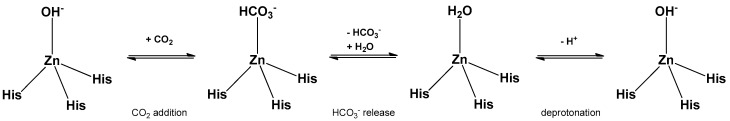
Proposed mechanistic processes for the catalytic cycle of CA.

**Figure 5 biomimetics-04-00066-f005:**
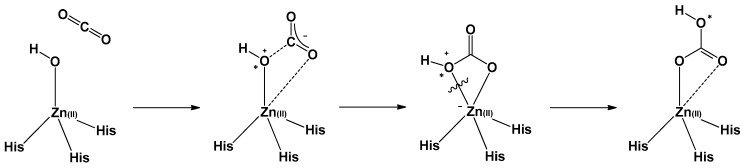
Mechanism for the formation of the bicarbonate intermediate, proposed by Lindskog.

**Figure 6 biomimetics-04-00066-f006:**
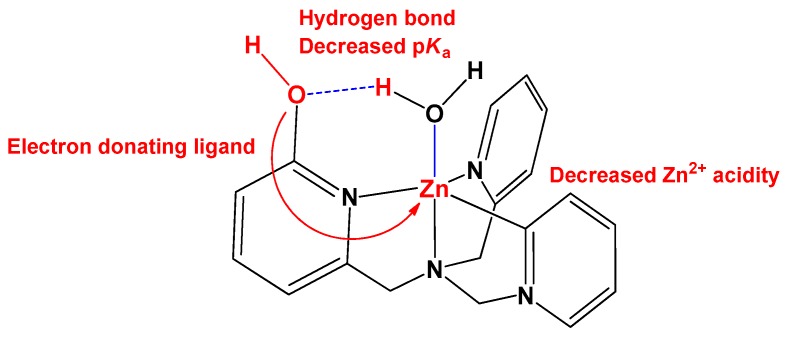
Schematic illustration of various effects by hydrogen bond mimicking.

**Table 1 biomimetics-04-00066-t001:** Observed p*K*_a_ of complexes (**1**) and (**2**).

Complexes	(1)	(2)
p*K*_a_	8.0	6.8

**Table 2 biomimetics-04-00066-t002:** Experimentally observed rate constants *k*_obs_, pH-independent *k*_ind_ rate constants, and standard deviations.

Complexes	*k* _obs_	*k* _ind_	σ_obs_
(**1**)	648.4	717.88	23.7
(**2**)	730.6	735.21	53.7

units are M^−1^s^−1^.

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
