# Peer review of "Kinetic Study of CO2 Hydration by Small-Molecule Catalysts with A Second Coordination Sphere that Mimic the Effect of the Thr-199 Residue of Carbonic Anhydrase"

_biomimetics, 2019, doi:10.3390/biomimetics4040066_

Round 1

Reviewer 1 Report

The manuscript describes the catalytic activity measurements of two molecular catalysts that mimic an enzyme for the hydration of CO2. The authors reported their unsuccessful attempts to achieve similar and better enzymatic activity with their systems and provided possible explanations. Below are detailed comments:

1. The manuscript mostly reports the final results and conclusion of the experiments. Many assumptions and understandings for how the experiments were conducted and why models were chosen for the analysis were implicitly implied. To improve the manuscript, the authors should explain these items in more details either in the manuscript or in the supporting information.  For example, why the author chose a certain indicator over another and why the absorbance was measured at particular wavelength. Also, the authors seem to assume that there the reaction is of first order. If so, they should state that and provide justifications.

2. Background of other similar molecular catalysts for this proposed reaction is lacking. What are the activity of those catalysts and how do them compare with the reported results?

3. What are the turnover rate and turnover numbers of the two reported molecular catalysts? How do they compare with those of the enzyme?

4. In terms of data display, it would be beneficial for the authors to display typical examples of their kinetic data than reporting the final analyzed results. 

5. The manuscript should state how the authors determine the Q factor.

6. The derivation for section 4 of the supporting information is not clear. The authors should present clearly how they obtained the last equation and how it was used for the data analysis.

Reviewer 2 Report

This paper can be acceptable in this journal after reviewing the following points:

 - Which are the advantages of using CA? What about economics?

 - Materials synthesis section needs to be re-writen. The scheme is not attractive for the reader and is difficult to follow.

 - Results section need to be deeper discussed. Please compare your results with other relevant works and highlight your novel findings.

Reviewer 3 Report

The work is devoted to the actual topic of carbonic anhydrase (CA) mimicking Zn-based complexes study. It presents new data on structure activity relation for CO2 hydration reaction I think the work could be published in Biomimetics after minor queries will be answered:

1) Introduction. CA activity is presented in terms of turnover number (why not turnover frequency which is better to compare reaction rate?), while activity of CA model catalysts I spresented in M^-1s^-1 units. This is inconsistent.

2) Last paragraph of Section 3.2. It would be great to support the discussion on complex structures by scheme or image.

3) What about mass-transfer influence on the kinetic measurements?

Round 2

Reviewer 1 Report

The authors replied to the reviewer's comments. However, they did not fully address the comments. Particularly, though their comments mentioned about adding some changes to the manuscript, but some of the proposed changes were not incorporated in the revised manuscript. This is frustrating to reviewer. Below are three examples:

1. One of the previous comments mentioned about errors in the equation (3) in section 5 of the supplementary information. Specifically, k(obs) was missing in the derivation. But this error is persistent in the last and current revision. As the authors pointed out their verbatim of the derivation in the article (Inorganic Chemistry. 51 (2012) 6803–6812), they did not even check if what they rephrased were exactly correct or expand upon the missing steps in the derivation.

2. The authors' reply mentioned about add some samples of detailed rate data in the manuscript but there were none. Only the final results were reported. 

3. The last review asked for turnover numbers and turnover frequency information about the catalysts. However, the authors replied information about k(obs) which is not the turnover number or the turnover frequency information. If the authors do not want to reply to this question, they should directly address it and give reasons than providing irrelevant information.

Author Response

We have made a major revision to the reviewer's comment to fully explain it.

1. The description of kobs is given in Equation (1) and Equation (3) in Section 2.3, and described in the supplementary information section 3 without omission in deriving the relationship between kind and kobs.

2. Section 3.3 has been completely revised to allow for a detailed discussion beyond just reporting on the rate of the samples.

3. The rate of CO2 hydration was measured by the pH indicator method using a stopped flow spectrometer. The rate equation for this can be expressed from equation (1) to equation (3). Equation (2) shows the relationship between the absorbance change of the stopped flow spectrometer and the CO2 hydration rate. The kobs values can be obtained from equations (2) and (3), which are discussed fully in section 2.3.

Reviewer 2 Report

Dear authors,

In my long career as research, is the first time that a response letter to reviewer that not contains the information that was changed or replaced. Your responses suggest that you do not like my comments. Moreover, not a thank you for taking time to read carefully your work or making comments to improve it.

You had your oppotunity to improve your work and you did not do properly. Therefore, I should recommend the rejection of your work. Nevertheless, I think I should provide another opportunity to response properly and therefore I would like you to re-write again the responses letter to review comments.

Best regards.

Author Response

Based on the reviewer's comments, we have made major revisions.

In particular, we tried to discuss in depth the pKa values in section 3.2 and the CO2 hydration rate results in 3.3. The revised key point is the discussion of acidity changes on Zn due to the formation of hydrogen bonding networks around Zn bound water, and therefore on pKa and rate changes.